# Sesamin Activates Nrf2/Cnc-Dependent Transcription in the Absence of Oxidative Stress in *Drosophila* Adult Brains

**DOI:** 10.3390/antiox10060924

**Published:** 2021-06-07

**Authors:** Tuan Dat Le, Yoshihiro H. Inoue

**Affiliations:** Insect Biomedical Research Center, Kyoto Institute of Technology, Matsugasaki, Sakyo-ku, Kyoto 606-8585, Japan; ltdat2662@gmail.com

**Keywords:** sesamin, *Drosophila*, anti-aging effects, brain, ARE, Nrf2/Cnc, neurons

## Abstract

Sesamin, a major lignin in sesame seeds, possesses health-promoting properties. Sesamin feeding suppresses several aging-related phenotypes such as age-dependent accumulation of damaged proteins in the muscles and neuronal loss in the brains of *Drosophila* adults with high levels of reactive oxygen species. Sesamin promotes the transcription of several genes that are responsible for oxidative stress, although the underlying mechanism remains unclear. Here, we aimed to demonstrate that sesamin mediates its action through activation of a transcription factor, Nrf2 (Cnc in *Drosophila*), essential for anti-aging oxidative stress response. Nrf2/Cnc activation was determined using the antioxidant response element, Green Fluorescence Protein reporter, that can monitor Nrf2/Cnc-dependent transcription. We observed strong fluorescence in the entire bodies, particularly in the abdomens and brains, of adult flies fed sesamin. Interestingly, Nrf2/Cnc was strongly activated in neuronal cells, especially in several neuron types, including glutamatergic and cholinergic, and some dopaminergic and/or serotonergic neurons but not in GABAergic neurons or the mushroom bodies of flies fed sesamin. These results indicate that the anti-aging effects of sesamin are exerted via activation of Nrf2/Cnc-dependent transcription to circumvent oxidative stress accumulation in several types of neurons of adult brains. Sesamin could be explored as a potential dietary supplement for preventing neurodegeneration associated with accumulation of oxidative stress.

## 1. Introduction

Sesame seeds have been used as a traditional health food supplement, as well as medicinal remedy [1,2]. They contain lignans that can significantly reduce various health risks [3,4,5,6,7,8]. Sesamin is the most abundant lignan in sesame seeds, and the catechol moiety of the sesamin molecule is responsible for the antioxidant and anti-inflammatory properties of sesame in mammals [9,10,11,12]. Sesamin restores tissue dysfunction induced by severe oxidative damage in many organisms [13,14,15,16,17]. Recent studies have shown that sesamin possesses antioxidative properties that can delay the appearance of aging-related phenotypes in the muscles, neurons, and intestinal stem cells of *Drosophila* [14]. However, the underlying mechanism is not clear.

Many research groups have performed screenings to identify new medicines and natural compounds that possess antioxidative properties similar to those of sesamin [1,18,19]. An evolutionarily conserved protein, nuclear factor erythroid 2–related factor 2 (Nrf2), has been identified as a prominent target of chemicals that possess antioxidative properties [20,21]. Nrf2 plays an important role in treating environmental oxidative stresses by upregulating enzymes for detoxification and antioxidation in various organisms [22,23]. It functions as a factor that induces transcription of target genes harboring antioxidant response elements (AREs) [20,24]. Kelch-like ECH-associated protein 1 (Keap1), an adaptor subunit of cullin 3-based E3 ubiquitin ligase, complexes with Nrf2 and stimulates proteasome-dependent degradation of the transcription factor [25]. Depending on cellular redox balance, Nrf2 is released from Keap1 upon recognizing oxidative and electrophilic signals, following which it escapes proteasomal degradation. The transcription factor moves to the nucleus, binds to ARE-driven gene promoters, and transactivates the expression of numerous cytoprotective genes [20,23,25]. Therefore, the Keap1-Nrf2-ARE axis plays a critical role as a significant signal in protecting cells from endogenous and exogenous oxidative stresses [24,25]. This signaling is considered to be an essential defense mechanism against the detrimental effects of oxidative stress in various diseases related to neurodegeneration, chronic inflammation, aging, and in cancer [22,23,26,27]. Recently, Nrf2 attracted attention as a prominent target for drug discovery for neurodegenerative diseases [28]. 

*Drosophila melanogaster* has been used as the most useful genetic model owing to its high fecundity and short life cycle. Furthermore, the availability of advanced genetic techniques has facilitated complicated experiments that investigate the effects of chemicals in the entire organism. *Drosophila* also has numerous advantages in drug discovery, particularly as a model for studies on aging [29,30]. We have characterized a hypomorphic mutant of *Sod1* encoding the Cu/Zn superoxide dismutase, which eliminates superoxide radicals [31]. Using these fly stocks, we have established an aging accelerated model that allows us to analyze the following aging-related phenotypes at early adult stages. As flies age, they display impaired locomotor activity, accumulation of abnormal protein aggregates containing polyubiquitinated proteins in the muscle, and loss of dopaminergic neurons in the brain [14,31]. By examining these aging-related phenotypes, it is possible to estimate the progress of aging in these tissues.

In this study, we investigated whether sesamin exerted an antioxidant effect on neurons in adult flies. We selected the following five types of neurons known to play critical physiological roles in mammalian brains: cholinergic neurons involved in wakefulness, the circadian system, and cognitive function; glutamatergic neurons involved in cognitive function, including motivation, reward association, and habit learning [32,33,34]; dopaminergic neurons involved in learning and memory [35]; serotonergic neurons involved in anxiety, sleep, appetite, and gastrointestinal motility [36]; and GABAergic neurons that act primarily as inhibitors at receptors in the adult vertebrate [32]. We also investigated whether sesamin exhibits antioxidant effects via stimulation of transcription mediated by Nrf2 (Cnc in *Drosophila*). Toward this, we used a GFP reporter to monitor the Nrf2/Cnc-dependent transcription of antioxidant genes [24]. In addition, we observed it’s effect on the mushroom body, which is a major memory center in *Drosophila* [37]. Overall, we addressed whether the effects of sesamin on activation of Nrf2/Cnc in the brain were for specific neurons or for various types of neurons. Furthermore, we investigated whether the effects of sesamin on activation of Nrf2/Cnc-dependent transcription resulted in suppression of loss of neurons due to the accumulation of oxidative stress. These genetic results indicated that the anti-aging effects of sesamin are exerted via activation of Nrf2/Cnc-dependent transcription to circumvent oxidative stress accumulation in several types of neurons of adult brains. Sesamin could be used as a dietary supplement for preventing neurodegeneration associated with the accumulation of oxidative stress.

## 2. Materials and Methods

### 2.1. Fly Stocks and Culture 

*w*^1118^ was used as the standard control stock. To monitor ARE-dependent transcription, the ARE-GFP reporter (ARE-GFP), in which a GFP cDNA was cloned after ARE sequences, was used [24]. A stock harboring ARE-GFP was a gift from Dirk Bohmann (University of Rochester Medical Center, Rochester, New York, NY, USA). The Gal4/UAS system was used for ectopic gene expression in *Drosophila* [38,39]. The following fly stocks were obtained from the Bloomington *Drosophila* Stock Center (Bloomington, IN, USA). For ubiquitous expression at more substantial and intermediate levels, *P{Act5C-GAL4}17bFO1* (*Act-Gal4*) (#81890) and *P{GAL4-arm.S}4a* (*Arm-Gal4*) (#1561), respectively, were used. The UAS-RNAi stock, *P{TRiP.JF02006}attP2* (*cncRNAi*) (#25984), was used for depletion of *cnc*. *P{EPgy2}Keap1[EY02632]* (*Keap1^EY^*) (#15427) was used for overexpression of *Keap1*. To visualize specific types of neuronal cells in adult brains, the following Gal4-drivers were used: *P{GAL4-elav.L}CG16779[3]* (*elav-Gal4*) (#8760) for expression in the nervous system, *P{VGlut-GAL4.D}1* (*VGlut-Gal4*) (#24635) for expression in glutamatergic neurons, *P{ChAT-GAL4.7.4}19B* (*ChAT-Gal4*) (#6798) for expression in cholinergic neurons, *P{ple-GAL4.F}3* (*TH-Gal4*) (#8848) for expression in dopaminergic neurons, *P{Ddc-GAL4.L}Lmpt[4.36]* (*Ddc-Gal4*) (#7009) for expression in serotonergic and dopaminergic neurons, *P{Gad1-GAL4.3.098}2* (*Gad1-Gal4*) (#51630) for expression in GABAergic neurons, and *P{GawB}4G* (*4G-Gal4*) (#6927) for expression in the mushroom body. The *P{UAS-tdTom.S}3* (*RFP*) (#36328) stock was used for labeling the neurons by Red Fluorescence Protein (RFP) expression. UAS-RNAi stocks, which can induce expression of dsRNAs against *Sod*1 mRNA [*P{UAS-Sod*1*.IR}4* (*Sod1RNAi*) (#24491)] and *Sod2* mRNA [*P{UAS-Sod2*,*dsRNA.K}15* (*Sod2RNAi*) (#24489)] were used for depletion of *Sod1* and *Sod2*, respectively.

Standard cornmeal diet, which contained 7.2 g agar, 100 g glucose, 40 g dried yeast, and 40 g cornmeal per liter was used as fly food. All ingredients were mixed well and boiled for 10 min. After cooling to 75 °C, 5 mL of 10% parahydroxybenzonate dissolved in ethanol and 5 mL of propionic acid were added to the diet. All fly stocks were maintained on a regular cornmeal diet at 25 °C, with the exception of depletion and overexpression experiments that were performed at 28 °C.

### 2.2. Chemical Feeding

Young adults were collected at two days after eclosion and separated into males and females. Twenty flies were reared in a single plastic vial containing *Drosophila* instant medium (Formula 4-24^®^ Instant *Drosophila* Medium, Blue; Carolina Biological Supply Company, Burlington, NC, USA). The flies were reared on the instant medium containing 10 mM Paraquat (Methyl viologen dichloride hydrate; Sigma-Aldrich, St. Louis, MO, USA). The flies reared on the instant medium without the chemical were used as controls. These flies were transferred into new vials with the diet every 2 days, and they were kept at 25 °C. For sesamin feeding, 2-day-old young adults were collected and separated into males and females. Twenty flies were reared in a single plastic vial containing the *Drosophila* instant medium. Sesamin (Nacalai Tesque, Kyoto, Japan) dissolved in 0.5% DMSO (Dimethyl sulfoxide; Wako Pure Chemical Industries, Ltd., Osaka, Japan) was added to final concentrations of 2 mg/mL in the instant food. As a control, 0.5% DMSO alone was added to the instant food. To feed the flies on the instant medium with the chemicals at 25 °C, the vials with the diet were changed to fresh ones after every 2 days. To raise the flies at 28 °C and induce ectopic expression via the UAS-Gal4 system, food vials were changed after every 1−2 days.

### 2.3. Observation of GFP and RFP Fluorescence

To visualize GFP fluorescence in the whole bodies, we collected ARE-GFP flies within 2 days after eclosion and fed them with 2 mg/mL sesamin (0.5% DMSO alone as a control) for 7 days or with 0%, 0.1%, 0.5%, 1%, and 2% DMSO for 7, 20, and 30 days. Then, the GFP fluorescence was observed using a stereo fluorescence microscope (SZX7; Olympus, Tokyo, Japan). To visualize GFP and/or RFP fluorescence in the brains or gut, flies harboring ARE-GFP were collected within 2 days after eclosion and fed 2 mg/mL sesamin (0.5% DMSO alone was used as a control). To visualize neurons in adult brains, the flies carrying neuron-specific Gal4 drivers were crossed with those with *UAS-RFP*. ARE-GFP flies were collected 2 days after eclosion and raised for 7 days on a diets supplemented with 2 mg/mL sesamin or 0.5% DMSO alone as a control. Brain or gut samples were dissected from the adults as described previously [14,31]. The whole brains or gut were fixed in 4% paraformaldehyde for 30 min, washed three times with PBST (phosphate-buffered saline containing 0.1% Triton X-100) for 10 min, and subsequently blocked with 10% normal goat serum for 30 min, followed by being washed three times in PBST for 10 min. All samples were mounted in Vectashield (Vector Laboratories, Burlingame, CA, USA). The GFP and/or RFP fluorescence was observed using an Olympus laser scanning confocal microscope (Fv10i; Olympus, Tokyo, Japan). The brightness and contrast of the entire images were adjusted using the Fv10i software.

### 2.4. Quantitative Reverse Transcription Polymerase Chain Reaction

For quantitative reverse transcription polymerase chain reaction (qRT-PCR) analysis, total RNA was extracted using TRIzol reagent (Invitrogen, Carlsbad, CA, USA) from the whole bodies, heads, and abdomens of adult flies fed diets with or without 2 mg/mL sesamin for 7 days. cDNA synthesis was performed using a PrimeScript II High Fidelity RT-PCR kit (Takara, Shiga, Japan) with oligo dT primers. qRT-PCR was performed using FastStart Essential DNA Green master mix (Roche, Mannheim, Germany) and a LightCycler Nano (Roche, Basel, Switzerland). *RP49* was used as the normalization reference. Relative mRNA levels were quantified using the LightCycler Nano software version 1.0 (Roche, Basel, Switzerland). The primers used were as follows:
RP49-Fw, 5′-TTCCTGGTGCACAACGTG-3′,RP49-Rv, 5′-TCTCCTTGCGCTTCTTGG-3′,GFP-Fw, 5′-AAGCTGACCCTGAAGTTCATCTGC-3′,GFP-Rv, 5′-CTTGTAGTTGCCGTCGTCCTTGAA-3′,Nrf2-Fw, 5′-TTACATCTACGAGTACGCCGC-3′,Nrf2-Rv, 5′-ACTGGAGCTCAAAACCGCTAA-3′,Keap1-Fw, 5′-CCACCGTGGAGCGTTATGATA-3′,Keap1-Rv, 5′-TTCCTGCATTCTGGACCAAGG-3′


All qRT-PCR experiments were performed in triplicate, and an average of three replicates in each group were considered. The ∆∆Ct method was used to determine the differences in target gene expression relative to the reference *Rp49* gene expression [40].

### 2.5. Quantitation of GFP or RFP-Positive Area

To measure GFP fluorescence and quantitate the GFP-positive area in the whole bodies, heads, or abdomens, fluorescence images were acquired using a stereo fluorescence microscope, and the images were analyzed using the Image J software (National Institutes of Health, Bethesda, MD, USA). The intensity or area of GFP or RFP fluorescence in the brain and gut cells were measured using the Fv10i software and counted using the Image J software. The quantification was also performed using the Image J software.

### 2.6. Statistical Analysis

Statistical analyses were performed using GraphPad Prism (Version 9, GraphPad Software, San Diego, CA, USA). The Student’s *t*-test was used for comparing the two groups. One-way analysis of variance (ANOVA) was used for analyzing differences in more than two groups. *p*-values < 0.05 were considered statistically significant.

## 3. Results

### 3.1. Sesamin Feeding Stimulated Expression of the ARE-GFP Reporter That Monitors Nrf2/Cnc-Dependent Transcription without External Oxidative Stress

Previously, we have shown that sesamin suppresses several aging phenotypes accelerated by oxidative stress in adult muscles, neurons, and gut. Simultaneously, sesamin feeding increased the mRNA levels of several antioxidative genes [14]. Based on results, we hypothesized that the Nrf2/Cnc-dependent antioxidative response was involved in the anti-aging and antioxidative effects of sesamin in *Drosophila*. Under oxidative stress, Nrf2 is activated and promotes transcription from specific DNA sequences called the ARE [20]. Therefore, we analyzed whether sesamin can activate Nrf2/Cnc-dependent transcription from the ARE in adult flies. To perform this, we used an ARE-GFP reporter that consists of the cDNA encoding GFP after the ARE sequences [24]. Before performing the sesamin feeding experiments, we confirmed that the expression of the reporter was stimulated in adults after feeding on a well-known oxidant, paraquat (Figure 1A,B). Furthermore, compared to the flies fed only on a fly diet, we observed a significant increase in the relative mRNA levels of *GFP* mRNA in the whole bodies of flies after paraquat feeding (Figure 1C). These results demonstrated the activation of a *Drosophila* Nrf2 orthologue, Cnc, in response to oxidative stress using the ARE-GFP reporter.

Next, we investigated whether sesamin can activate the expression of the ARE-GFP reporter. Sesamin is a water-insoluble chemical but can be easily dissolved in DMSO. DMSO exerts a deleterious effect when used at high concentrations as a solvent to dissolve water-insoluble chemicals [41]. Therefore, initially, we determined the concentration of DMSO that exerted minimum effect on ARE-GFP expression. We fed ARE-GFP flies (2-day old) a diet containing five different concentrations of DMSO including 0% (as a control), 0.1%, 0.5%, 1.0%, and 2.0% for 7, 20, and 30 days. GFP fluorescence was observed in the ARE-GFP flies fed a diet with 1.0% and 2.0% DMSO continuously for 7, 20, and 30 days (Appendix A). In contrast, we did not observe any significant increase in ARE-GFP fluorescence in flies fed 0.1% and 0.5% DMSO from 7 to 20 days. As it was considerably difficult to dissolve 2 mg/mL sesamin in 0.1% DMSO, we decided to use 0.5% DMSO to dissolve sesamin in the fly diet. The adults showed a level of *GFP* mRNA that was five times higher in males and twenty times higher in females on the first day after eclosion without sesamin administration, compared to the levels in 3-day old flies (Appendix A). The mRNA level decreased the following day and was maintained at a constant and low level thereafter. Subsequently, we performed the feeding experiments using 2-day-old flies after eclosion. We selected 2-day-old adult flies harboring ARE-GFP and fed them a fly diet supplemented with 2 mg/mL sesamin in 0.5% DMSO or a diet with 0.5% DMSO (as a control) for 7 days. Interestingly, stronger GFP fluorescence was observed in whole bodies of the adults fed sesamin than in the adults fed a diet containing DMSO alone (Figure 1D,E). After sesamin feeding, *GFP* mRNA levels increased by 101% in females fed sesamin as compared to the females in the control group that were fed DMSO. Consistently, the levels increased by 39% in sesamin-fed males compared to the males in the control group (Figure 1F). Therefore, we concluded that the expression of the ARE-GFP reporter was activated in flies upon continuous feeding of sesamin in the absence of excess oxidative stress.

### 3.2. Sesamin Feeding Stimulated the ARE-GFP Expression in Adult Brain and Gut

After sesamin feeding, we observed distinct GFP fluorescence in the head and abdomen. These observations are consistent with previous results showing that sesamin suppressed progression of the aging phenotypes that appeared in the adult nervous system and intestine [14]. Cnc, the *Drosophila* Nrf2 orthologue, plays a vital role in mitigating oxidative damages [24]. Therefore, we focused on assessing the effect of sesamin on Nrf2/Cnc-mediated transcription in adult brains and gut. Using confocal microscopy, we observed stronger GFP fluorescence in the brains of both adult male and female sesamin-fed (Figure 1G–I) flies with ARE-GFP than in those of control flies. Particularly, we observed strong GFP fluorescence in the central region of the brain (Figure 1G2,G4,H). Furthermore, we quantitated the relative *GFP* mRNA levels using total RNA prepared from fly heads via qRT-PCR. The feedings increased the mRNA levels by 95% in males and by 34% in females (Figure 1I). Consistently, we also observed considerably stronger GFP fluorescence in the foregut (especially in crop) and hindgut and moderately stronger fluorescence in the midgut of adults fed sesamin than in those of control flies (Figure 1J1–J4,K). qRT-PCR for assessing *GFP* mRNA level in the abdomens indicated that sesamin feeding increased ARE-GFP expression by 58% in males and by more than 222% in females (Figure 1L).

### 3.3. Sesamin Activated the Expression of the ARE-GFP Reporter in Adult Brains in a Nrf2/Cnc-Dependent Manner

We next investigated whether sesamin-induced activation of *ARE-GFP* expression in the brain depended on Cnc. We confirmed that dsRNA against *cnc* mRNA induced using *UAS*-*cncRNAi* and *Act-Gal4* for more severe depletion in whole bodies, or *Arm-Gal4* for less severe but ubiquitous depletion efficiently depleted the endogenous *cnc* mRNA (Appendix A). Subsequently, we investigated whether the effects of sesamin disappeared in the *cnc*-depleted brains (Figure 2A). We did not observe significant activation of ARE-GFP expression in the brains of *cnc*-depleted males and females after sesamin feeding (Figure 2B,C). These results indicated that ARE-GFP expression was stimulated especially in the brain, which was dependent on Nrf2/Cnc. 

### 3.4. Overexpression of Keap1 Abolished Induction of ARE-GFP Expression by Sesamin

Keap1 binds to Nrf2 and stimulates its proteasome-dependent degradation [27]. Therefore, we investigated whether overexpression of *Keap1* abolished the sesamin-induced ARE-GFP expression. We observed GFP fluorescence in the adult brains harboring neuron-specific overexpression of *Keap1* from adults (*elav* > *Keap1^EY^*) raised at 28 °C for 7 days on the diet with sesamin (Figure 2D). After sesamin feeding, weaker GFP fluorescence was observed in both males (Figure 2D4,E) and females (Figure 2D8,E), whereas the fluorescence almost disappeared in male and female brains overexpressing *Keap1* (Figure 2D3,D7,E) without sesamin. Using qRT-PCR, we quantified *GFP* mRNA levels in the heads of sesamin-fed adult flies (Figure 2F). We confirmed that expression of the ARE-GFP reporter in adult brains induced by sesamin feeding was abolished by depletion of *cnc* as well as *Keap1* overexpression. These results confirmed that sesamin activated ARE-dependent transcription is regulated by Keap1-Nrf2/Cnc signaling in the brain.

### 3.5. Sesamin Feeding Stimulated Nrf2-Dependent Transcription in Several Types of Neurons Present in Adult Brains

Previously, we have shown that sesamin partially suppressed oxidative stress- and age-dependent loss of dopaminergic neurons [14]. In this study, we first investigated whether sesamin activated Nrf2/Cnc-dependent transcription in adult brains using the ARE-GFP reporter. We observed strong GFP fluorescence in several areas in the central brain (Figure 1G). The mushroom body in the central brain acts as the central nervous system for olfactory learning and memory in *Drosophila* [37]. Therefore, we assessed whether sesamin activated Nrf2/Cnc-dependent transcription in specific neuronal subtypes, or whether it induced transcription in various neurons. For this purpose, we selected five types of neurons known to play critical physiological roles in human brains. We observed ARE-GFP expression in the cell bodies of each neuron type exclusively labeled via RFP fluorescence using a neuron type-specific Gal4 driver in *Drosophila* adult brains: *TH-Gal4* for dopaminergic neurons, *VGlut-Gal4* for glutamatergic neurons, *ChAT-Gal4* for cholinergic neurons, *Gad1-Gal4* for GABAergic neurons, and *Ddc-Gal4* for serotonergic and dopaminergic neurons. In addition, we observed reporter expression in the mushroom bodies using *4G-Gal4*. Using confocal microscopy, we examined whether the fluorescence of the ARE-GFP reporter in the cell body clusters of each neuron was visualized using RFP fluorescence. We fed the flies a diet containing 2 mg/mL sesamin for 7 days. We have previously demonstrated that sesamin feeding partially suppresses the loss of dopaminergic neurons in adult brains displaying reactive oxygen species (ROS) accumulation [14]. Consistently, we observed that areas showing expression of the ARE-GFP reporter in the areas occupied by dopaminergic neurons (*TH* > *RFP*) increased after sesamin feeding in both male (6% to 9%) and female brains (5% to 10%) (n >= 27 flies were examined) (Figure 3I,K). The areas expressing ARE-GFP in the total area of dopaminergic neurons significantly also increased by 81% for males and 52% for females after sesamin feeding (*p* < 0.05, Student’s *t*-test) (Figure 3J). Consistently, the total intensity of GFP fluorescence in the neuron areas also increased significantly after sesamin feeding in males (*p* < 0.05) and females (*p* < 0.01) (Student’s *t*-test) (Figure 3L). Therefore, sesamin feeding induced ARE-GFP expression, indicating activation of Nrf2/Cnc in the dopaminergic neurons of adult brains.

These observations prompted us to investigate whether sesamin also induced ARE-GFP expression in the other four types of neurons. The glutamatergic neuron area (*VGlut* > *RFP*) in the brains was 40% for males (6% in control brains without sesamin feeding) and 36% (4% in control) for females fed sesamin (Figure 3A,C). The total area expressing ARE-GFP in the neurons significantly increased by six-fold in the brains of both males and females after sesamin feeding (*p* < 0.0001, Student’s *t*-test) (Figure 3B). The intensity of GFP fluorescence also increased significantly after sesamin feeding in both sexes (Figure 3D). Furthermore, we observed a higher frequency of GFP fluorescence of the ARE-GFP reporter in cholinergic neurons; 23% in males on average (7% in controls not fed sesamin) and 15% in females on average (6% in controls fed without sesamin) (Figure 3E,G). The total area expressing ARE-GFP in the neurons significantly increased by three-fold in both male and female brains after sesamin feeding (*p* < 0.0001, Student’s *t*-test) (Figure 3F). The intensity of GFP fluorescence indicated that the expression levels in the neuron areas also increased significantly after sesamin feeding in both sexes (Figure 3H).

In contrast, in the serotonergic and dopaminergic neuron area (*Ddc* > *RFP* on average), slightly stronger fluorescence was detected in only a few neurons (surrounded by dotted lines) in males and females after sesamin feeding (Figure 3M–O). However, we did not detect GFP fluorescence of the ARE reporter over background level in the GABAergic neuron area (*Gad1* > *RFP* on average) (Figure 3P–R). Expression of ARE-GFP was not detected in the mushroom bodies (Figure 3S–U). Overall, these results indicated that sesamin feeding induced transcription mediated by Nrf2/Cnc without excess oxidative stress in several types of neurons in adult brains, particularly more efficiently in glutamatergic and cholinergic neurons than in dopaminergic and serotonergic neurons, but not in GABAergic neurons or mushroom bodies.

### 3.6. Sesamin Suppressed Reduction of Glutamatergic, Cholinergic, and Dopaminergic Neurons Associated with Oxidative Stress Accumulation Induced by Sod1 or Sod2 Depletion

Next, we assessed whether sesamin-induced activation of Nrf2/Cnc-mediated transcription suppressed loss of neurons under the condition of excess ROS accumulation. A previous study has demonstrated that sesamin feeding partially suppresses loss of dopaminergic neurons in adult brains with ROS accumulation [14]. Therefore, we first confirmed whether sesamin feeding suppressed the reduction of dopaminergic neurons associated with oxidative stress accumulation due to neuron-specific *Sod2* depletion. After dopaminergic neuron-specific depletion of *Sod2* in adult brains (*TH* > *RFP*, *Sod2RNAi*) (Appendix A), the total neuron area (4000 pixels on average) (Appendix A) in male brains reduced by 22% as compared to the areas without depletion (5100 pixels on average) (*TH* > *RFP*, Appendix A). The total neuron area (3900 pixels on average) (Appendix A) in female brains was 20% smaller than the areas without depletion (4900 pixels on average) (*TH* > *RFP*, Figure 3SB). After sesamin feeding, the total dopaminergic neuron areas increased significantly in both males (*p* < 0.05) and females (*p* < 0.001), as compared to the areas in adult brains without sesamin feeding. These observations are consistent with our previous findings that sesamin can partially suppress the loss of neurons in a few dopaminergic clusters in *Sod1*-depleted adult brains [14].

We next investigated whether sesamin suppressed the loss of other types of neurons. In both cholinergic and glutaminergic neurons, we observed stronger activation of Nrf2/Cnc after sesamin feeding than without feeding of the chemical. We selected these two types of neurons and examined whether sesamin feeding for 7 days affected the amount of each neuron. We also investigated whether sesamin suppressed the reduction of areas occupied by cholinergic neurons under conditions of oxidative stress accumulation. To achieve these conditions in the neurons, we depleted *Sod1* because of the ease of construction of fly stocks simultaneously harboring *UAS-Sod1RNAi* and *ChAT-Gal4* (Figure 4A). Next, we confirmed whether sesamin modulates a change in the number or morphology of cholinergic neurons under accumulation of excess oxidative stress due to *Sod1* depletion in both males and females. After cholinergic neuron-specific depletion of *Sod1* in adult brains (*ChAT* > *RFP*, *Sod1RNAi*), the total neuron area (2000 pixels on average in Figure 4C) in male brains reduced by 43% compared to the areas without depletion (3500 pixels Figure 4B). The total neuron area (1800 pixels on average in Figure 4C) in female brains was 36% smaller than the areas without depletion (2800 pixels on average in Figure 4B). Upon sesamin feeding, the total area of the neurons increased significantly in both males and females (*p* < 0.0001).

The total area of the glutaminergic neurons increased slightly in male brains after sesamin feeding, although we did not observe any statistically significant increase in total or average areas (*VGlut* > *RFP*) in male or female brains (Figure 5B). In adult brains harboring glutaminergic neuron-specific depletion of *Sod2* (*VGlut* > *RFP*, *Sod2RNAi*), the total neuron area (5000 pixels on average in Figure 5C) in male brains was 8% smaller than the areas without depletion (5400 pixels Figure 5B). The total area (3700 pixels on average in Figure 5C) in female brains was 12% smaller than the areas without depletion (4200 pixels in Figure 5B). Fewer glutaminergic neurons were maintained in female brains with neuron-specific accumulation of oxidative stress (*VGlut* > *RFP*, *Sod2RNAi*) than in control female brains (*VGlut* > *RFP*). However, after sesamin feeding, the total area of the neurons increased in both males and females harboring *Sod2RNAi* (*p* < 0.05, Figure 5B,C). Sesamin feeding also exerted less effective but consistent effects on the survival of glutaminergic neurons under oxidative stress. In summary, these results imply that sesamin exerts an antioxidative effect that suppresses the reduction of the areas occupied by dopaminergic, cholinergic, and glutaminergic neurons associated with neuron-specific accumulation of oxidative stress in *Drosophila* males and females without external oxidative stress.

## 4. Discussion

Although evidence regarding the antioxidative effects of sesamin are increasing, the mechanism underlying the effects of sesamin has not been completely understood. Our observations revealed that sesamin promoted activation of the transcription factor Nrf2/Cnc, which plays an important role in antioxidative response. Consistently, a study showed that sesamin significantly activated transcription of Nrf2 target genes in human colorectal adenocarcinoma cells, whereas depletion of Nrf2 abolished its protective effect [42,43].

In this study, we used the transgenic ARE-GFP reporter line (ARE-GFP) to monitor the ARE-dependent transcription mediated by the Nrf2/Cnc transcription factor in adult flies. Nrf2/Cnc is associated with Keap1, which is an inhibitor of Nrf2/Cnc in the absence of oxidative stress. The complex is decomposed via proteasome-mediated protein degradation. After the release of Keap1 from the complex, Nrf2/Cnc can initiate transcription from the ARE of each target gene [20,24]. Our observation of GFP fluorescence in the whole bodies, especially in the gut and brains, after sesamin feeding in the absence of any other oxidants, indicated that sesamin was responsible for the activation of Nrf2/Cnc in these adult tissues. In our previous study [14], we demonstrated that sesamin feeding raised mRNA levels of several antioxidative genes simultaneously, and the expression of *GstD1-GFP* reporter that can be induced dependent on ROS accumulation was suppressed after continuous sesamin feeding. We interpreted that the suppression is a consequence of the antioxidation effect of sesamin. By contrast, a previous study using mammalian PC12 cells reported that the intracellular ROS level transiently increased in the cultured cells after the addition of the sesamin metabolite, SC-1 [43]. The authors discussed that metabolite-induced ROS directly interacts with the SH groups of Keap1, and activates p38 which possibly phosphorylates Nrf2. The resultant Nrf2/ARE activation may induce expression of antioxidant genes. Although we do not have data to conclude whether the chemical temporally induces ROS in *Drosophila* bodies, it may be reasonable to consider that the antioxidation effect of sesamin appears in *Drosophila* via similar processes. Several recent studies have suggested that Keap1-Nrf2 is a promising target for the discovery of drugs that prevent various murine and human cancers [26,44]. Keap1 is a sulfhydryl-rich protein that represses Nrf2 signaling [45]. For example, interaction of exogenous supplements such as sulforaphane with Keap1 disrupts this function and releases Nrf2 from its inhibitor [44]. Subsequently, the transcription factor can translocate to the nucleus and activate the transcription of its target genes [21]. Another study reported that the α,β-unsaturated carbonyl system of chalcones can activate the transcription of Nrf2 target genes by disrupting the linkage between Keap1 and Nrf2 [25]. We showed that induction of the ARE-dependent expression by sesamin feeding almost disappeared in the brains of *cnc*-depleted flies. In contrast, the sesamin feeding still slightly enhanced the expression in the same tissues harboring *Keap1* overexpression, although the overexpression almost disrupted the ARE-dependent expression without sesamin feeding. Sesamin might activate transcription via inhibition of Keap1, and consequently, it may enhance Nrf2/Cnc-dependent transcription. The target of sesamin should be identified by investigating whether the chemical directly binds to Keap1.

We observed strong GFP fluorescence of the ARE-GFP reporter in the heads and abdomens of adult flies fed with sesamin, indicating that ARE-dependent transcription by Nrf2/Cnc was induced in specific tissues by sesamin feeding. The strongest activation of Nrf2/Cnc was observed in the foregut and hindgut, and moderately stronger expression was observed in the midgut. The *Drosophila* gut is structurally and functionally analogous to the human intestine [46], and it is divided into three regions: foregut, midgut, and hindgut. The anterior midgut and the crop in the foregut function as the stomach in *Drosophila*. The posterior midgut is the most metabolically active region of the gut and is analogous to the human small intestine, while the hindgut corresponds to the human colon [47]. Low ARE-expression was also found in the crop and hindgut after feeding the standard diet without sesamin. It is possible that gut epithelial cells possess an intrinsic regulatory system that protects the cells from oxidants consumed through the diet, and sesamin may activate the system more efficiently. Previously, we showed that sesamin is digested into its metabolite, SC1, in gut epithelial cells [14]. If we correlate or extrapolate the observations in *Drosophila* to humans, it would be interesting as a future perspective to investigate whether sesamin could activate the Nrf2-mediated antioxidant responsible system in epithelial cells in the mammalian stomach and small intestine. Thereafter, the metabolite may be absorbed in gut epithelial cells, exported in the hemolymph, and transported toward the adult brain. Subsequently, it acts on neurons in adult brains. If sesamin or its metabolites could cross the blood brain barrier (BBB) in *Drosophila*, as the *Drosophila* BBB is known to be more permeable than mammals [48], it could act on neurons in adult brains. Alternatively, the information of sesamin or its metabolite generated in epithelial cells may be transmitted toward the brain via a gut–brain neuronal circuit [49]. These interesting observations will be further verified in future. In other adult tissues, with the exception of the brain and gut, expression over background levels was not observed. Thus, the effects of sesamin on the transcription of target genes, controlled by the Keap1-Nrf2-ARE regulatory system, vary among adult tissues. The action mechanism of the chemical may differ among adult tissues.

Considerable enhancement of ARE-dependent transcription was observed in three of the five types of neurons studied, although we could not specify the clusters of dopaminergic neurons, in which the ARE-Cnc was activated by sesamin feeding in this time. We observed that sesamin feeding enhanced ARE-dependent transcription in glutamatergic neurons and cholinergic neurons distinctively and enhanced it in dopaminergic and serotonergic neurons less remarkably, although the underlying reason is currently unknown. However, sesamin feeding significantly suppressed neuronal loss and/or reduced gene expression due to neuron-specific ROS accumulation in dopaminergic, cholinergic, and glutaminergic neurons, in which we observed significant activation of transcription from ARE by Nrf2/Cnc. The suppression in the dopaminergic neurons is consistent with our previous finding that sesamin can partially suppress the loss of neurons observed in a few dopaminergic clusters in adult brains harboring *Sod1* depletion [14]. Several studies have reported that Nrf2 upregulation may prevent the reduction of neuronal action in neurodegenerative diseases [28,50]. Furthermore, other studies have also reported that genetic activation of Nrf2 was sufficient to ameliorate neurodegenerative phenotypes, including loss of dopaminergic neurons in *Drosophila* Parkinson’s disease models [51,52]. In this study, we observed that sesamin feeding activated Nrf2/Cnc in cholinergic and glutamatergic neurons more frequently and at higher levels than in other neurons. A cholinergic neuron provides the neurotransmitter acetylcholine to the cerebral cortex and promotes cortical activation during wakefulness and rapid eye movement sleep in humans [53]. The dysfunction and loss of cholinergic neurons in the fore brain are involved in neural degradation during mammalian aging or in Parkinson’s disease and Alzheimer’s disease models. Consistent with this, a recent *Drosophila* study reported that inhibition of oxidative stress in cholinergic projection neurons upon *Sod2* overexpression can completely rescue aging-associated olfactory circuit degeneration [54]. A glutamatergic neuron produces glutamate, which is one of the most common excitatory neurotransmitters in human brains. It also plays an important role in several processes, including learning, cognition, and memory. In *Drosophila*, the physiological effects of the neurons are largely unknown, although the neurons are abundant in the central nervous system. A recent study revealed that glutamate acts as an inhibitory neurotransmitter in the olfactory system [55]. Furthermore, previous in vitro studies using mammalian cultured cells showed that sesamin promoted neurite outgrowth under insufficient nerve growth factor conditions in rat pheochromocytoma cells [8]. Considering these previous findings, our observations suggest that sesamin might suppress age-dependent impairment of long-term memory. Further experiments to clarify this hypothesis in *Drosophila* and mammalian models are required. In contrast, we failed to detect ARE-dependent transcription in GABAergic neurons, which play a major inhibitory role in vertebrate brains [56]. Recent studies in *Drosophila* have shown that several GABAergic neurons act as critical brakes to prevent incessant feeding and inhibit consumption [57]. However, sesamin feeding failed to activate Nrf2-mediated transcription in these types of neurons or in the mushroom bodies under our feeding conditions. The mushroom body is a central brain structure necessary for olfactory memory and for the selectivity of learned responses to specific odors. It contains more than 2000 neurons, including 20 types of neurons that use the neurotransmitter dopamine [58]. Although we did not detect ARE-GFP expression in and around the mushroom body, further experiments involving continuous long-term administration of sesamin are required before reaching any conclusion. These results indicate that the effect of sesamin on the activation of Nrf2/Cnc varies between different neuron types. Furthermore, our current data suggest that sesamin feeding could suppress the loss of neurons and/or their morphological alterations upon oxidative stress. Another recent study demonstrated that constitutive overexpression of *cnc* suppresses impairment of synaptic function and longevity after exposure to stress. Consistently, removal of its inhibitor, Keap1, showed beneficial effects on synaptic function and longevity [59]. It is possible that the Keap1-Cnc signal contributes to the suppression of age-dependent impairment of neurons. We demonstrated that sesamin can initiate an antioxidative stress response in some types of neurons before accumulation of oxidative stress over a detectable level. Our previous study also showed that sesamin feeding extended the lifespan of adult flies and suppressed age-related phenotypes [14]. In this study, we investigated Nrf2/Cnc-mediated transcription using an ARE-GFP reporter. It is also important to confirm the observations by examining mRNA levels of known target genes of *cnc*. This is a limitation of the current study. We will present the results in our next manuscript. Whether sesamin shows a similar preventive effect that protects these two types of neurons against oxidative stress in mammalian neurodegeneration disease models and during their normal aging will be interesting to investigate.

## 5. Conclusions

This study demonstrated that sesamin protected *Drosophila* adults against oxidative damage via stimulation of the Nrf2/Cnc-dependent transcription in the adult gut and brain. Sesamin feeding also enhanced transcription in several neurons other than dopaminergic neurons, such as cholinergic and glutaminergic neurons, in adult brains. The feeding rescued loss of these neurons induced by neuron-specific depletion of *Sod* genes. These results suggest that sesamin can be used for preventing neurodegeneration associated with accumulation of oxidative stress.

## Figures and Tables

**Figure 1 antioxidants-10-00924-f001:**
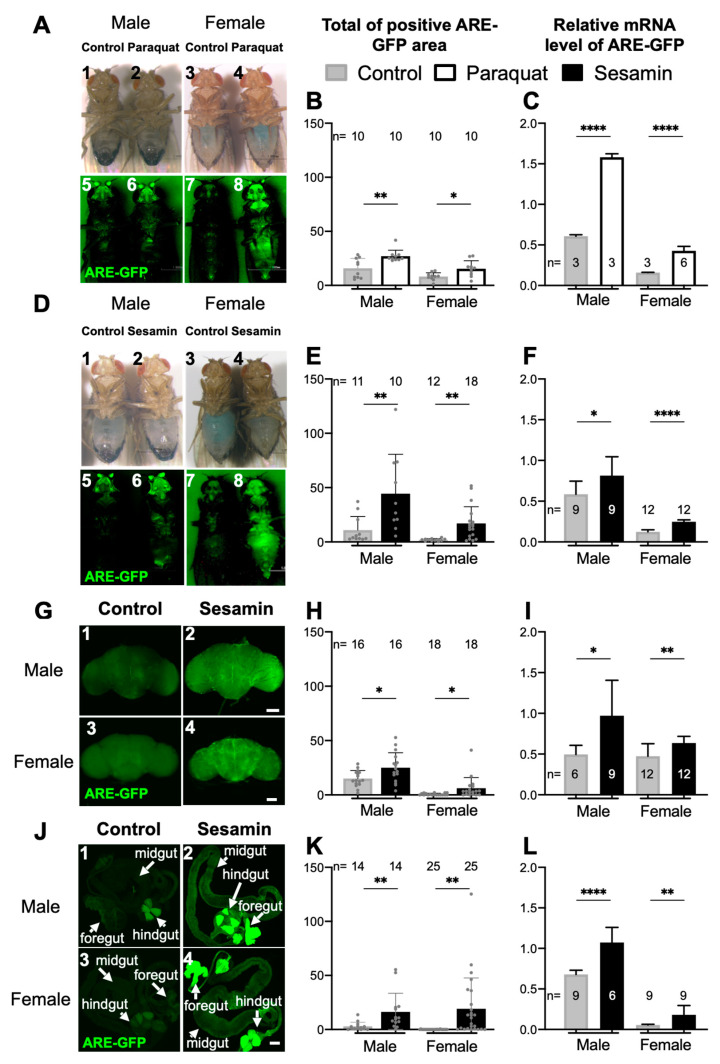
GFP fluorescence of antioxidant response element (ARE)-GFP reporter in whole bodies, brains and gut in adults fed sesamin. (**A**) Stereo micrographs of male and female flies with ARE-GFP reporter (ARE) fed on a fly diet supplemented with 10 mM paraquat (paraquat) for 7 days. (**A1**–**A4**) Bright field microscopy of a male (**A1**) and a female (**A3**) without paraquat feeding, and a male (**A2**) and a female (**A4**) with the paraquat administration. (**A5**–**A8**) GFP fluorescence after irradiation of excitation light for GFP. (**B**) Quantification of GFP positive area in whole bodies of males or females fed on the diet with or without paraquat. The area was calculated by Image J for each treatment (*n* = 10). (**C**) Quantification of *GFP* mRNA levels in males or females fed on the diet with or without paraquat. Paraquat feeding stimulates a transcription of ARE-GFP reporter. (**D**) Stereo fluorescence micrographs of male and female flies with ARE, fed with 2 mg/mL sesamin or without the chemical for 7 days. (**D1**–**D4**) bright field microscopy of a male (**D1**), and female (**D3**) without drug administration, and those of a male (**D2**) and a female (**D4**) with the sesamin administration. (**D5**–**D8**) GFP fluorescence after irradiation of excitation light for GFP. (**E**) Quantification of GFP positive area in the whole bodies of males or females fed on the diet with or without sesamin. The area was calculated by Image J for each treatment (*n* ≥ 10). (**F**) Quantification of *GFP* mRNA levels in the whole bodies of males or females fed on the diet with or without sesamin. The relative *GFP* mRNA level was quantitated by qRT-PCR performed using total RNA extracted from 10–15 whole bodies. The sesamin feeding 2 mg/mL stimulates transcription of ARE-GFP reporter in the whole adult bodies. (**G**,**J**) GFP fluorescence of confocal micrographs in adult whole brains (**G**), and adult gut (**J**). (**G1**–**G4**) GFP fluorescence of a male brain (**G1**), and female brain (**G3**) from adults raised without the chemical, and those of a male brain (**G2**) and a female brain (**G4**) from adults with the sesamin administration. (**H**) Quantification of GFP positive area in whole brains of males or females fed with or without sesamin. (**I**) The relative mRNA level of *GFP* was quantitated by qRT-PCR using total RNA from 40–50 heads. (**J**) Foregut, midgut, and hindgut are indicated individually by arrows. (**J1**–**J4**) GFP fluorescence of a male gut (**J1**), and female gut (**J3**) from adults raised without the chemical, and that of a male (**J2**), and female gut (**J4**) from adults with the sesamin administration. (**K**) Quantification of GFP positive area in gut of males or females fed with or without sesamin. (**L**) Quantification of *GFP* mRNA level in adult guts of males or females fed with or without sesamin. The relative mRNA level of *GFP* was quantitated by qRT-PCR using total RNA from 30–40 abdomens. The sesamin feeding stimulated ARE-dependent transcription in the adult brains and gut in both sexes. Scale bars in (**G**,**J**) represent 100 μm. ** p* < 0.05, *** p* < 0.01, ***** p* < 0.0001 (Student’s *t*-test).

**Figure 2 antioxidants-10-00924-f002:**
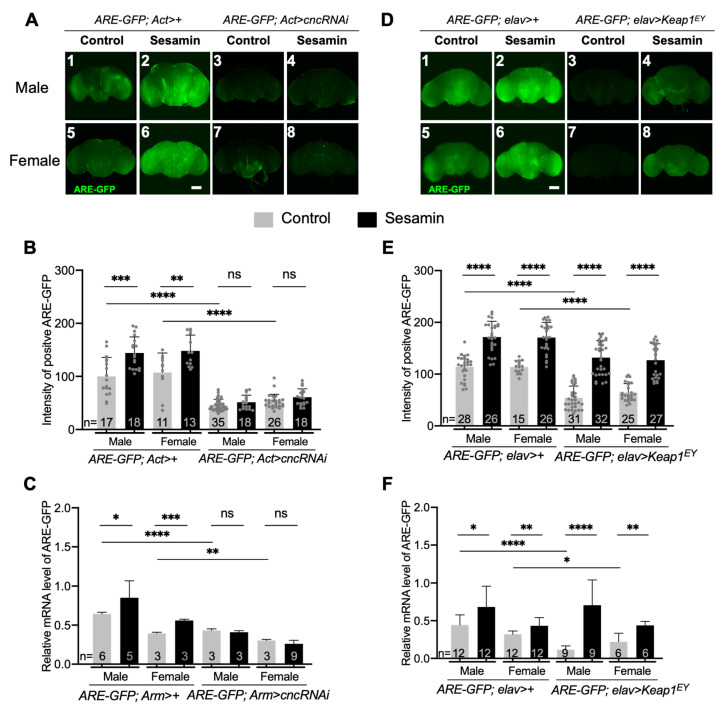
Reduction of sesamin-induced ARE-GFP expression by depletion of mRNA encoding a Nrf2/Cnc transcription factor or by ectopic overexpression of *Keap1*. (**A**,**D**) GFP fluorescence in brains from adults harboring ARE-GFP reporter (ARE). Brains were prepared from male flies (**A1**–**A4**,**D1**–**D4**) or female flies (**A5**–**A8**,**D5**–**D8**). (**A1**,**A2**,**A5**,**A6**) Brains from control flies with ARE-GFP (*ARE-GFP*; *Act*>*+*) fed on fly diet (**A1**,**A5**) or the diet supplemented with 2 mg/mL sesamin (**A2**,**A6**). (**A3**,**A4**,**A7**,**A8**) Brains from flies harboring depletion of *cnc* in all brain cells using Gal4 driver, Act-Gal4 and ARE-GFP (*ARE-GFP*; *Act* > *cncRNAi*) fed on the fly diet (**A3**,**A7**), or the diet with 2 mg/mL sesamin (**A4**,**A8**). (**D1**,**D2**,**D4**,**D5**) Brains from control flies with ARE-GFP (*ARE-GFP*; *elav* >*+*) fed on fly diet (**D1**,**D5**) or the diet supplemented with 2 mg/mL sesamin (**D2**,**D6**). (**D3**,**D4**,**D7**,**D8**) Brains from flies harboring ectopic overexpression of *Keap1* encoding an inhibitor of Nrf2/Cnc using a pan-neuronal Gal4 driver, *elav-Gal4* and *ARE-GFP* (*ARE-GFP*; *elav* > *Keap1^EY^*) fed on the diet (**Dc**3,**D7**), or the diet with 2 mg/mL sesamin (**D4**,**D8**). (**B**,**E**) Quantification of GFP intensity of brains from adults harboring ARE-GFP, with or without ubiquitous depletion of *cnc* (*ARE-GFP*; *Act*>*+*, *ARE-GFP*; *Act* > *cncRNAi*) (**B**), or brains harboring ARE-GFP with or without pan-neuronal expression of *Keap1* from adults (*ARE-GFP*; *elav*>*+*, *ARE-GFP*; *elav* > *Keap1^EY^*) (**E**), fed with or without 2 mg/mL sesamin. The intensity of the ARE-GFP fluorescence in adult brains (*n* ≥ 11) was calculated by Image J for each condition. (**C**,**F**) Quantification of *GFP* mRNA of brains from adults harboring ARE-GFP with or without ubiquitous depletion of *cnc* (*ARE-GFP*; *Act*>*+*, *ARE-GFP*; *Act* > *cncRNAi*) (**C**), or brains harboring ARE-GFP with or without pan-neuronal expression of *Keap1* (*ARE-GFP*; *elav*>*+*, *ARE-GFP*; *elav* > *Keap1^EY^*) (**F**), fed with or without 2 mg/mL sesamin. The relative *GFP* mRNA level was quantitated by qRT-PCR using total RNA from 40–50 heads for each condition. Scale bars represent 100 μm in the adult brain. ns not significant, ** p* < 0.05, *** p* < 0.01, **** p* < 0.001, ***** p* < 0.0001, Student’s *t*-test.

**Figure 3 antioxidants-10-00924-f003:**
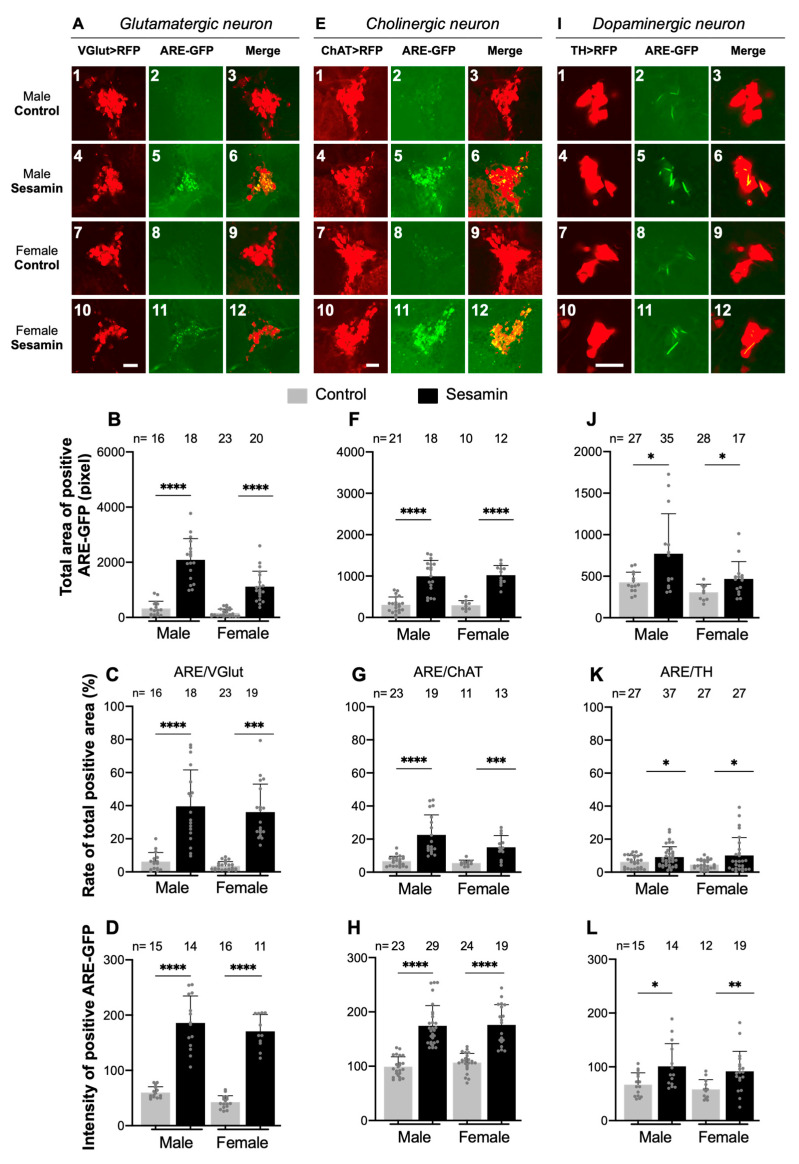
Expression of ARE-dependent transcription by Nrf2/Cnc in several types of neurons in the central adult brain. (**A**,**E**,**I**,**M**,**P**,**S**) Confocal microscopy observation of GFP fluorescence monitoring ARE-dependent transcription by Nrf2/Cnc transcription factor in five types of neurons and mushroom bodies in brains from adults fed on sesamin. The glutamatergic neurons (**A**–**D**), cholinergic neurons (**E**–**H**), dopaminergic neurons (**I**–**L**), serotonergic and dopaminergic neurons (**M**–**O**), GABAnergic neurons (**P**–**R**), and mushroom bodies (**S**–**U**) were labelled by RFP using VGlut-Gal4 (*VGlut* > *RFP*), ChAT-Gal4 (*ChAT* > *RFP*), TH-Gal4 (*TH* > *RFP*), Ddc-Gal4 (*Ddc* > *RFP*), Gad1-Gal4 (*Gad1* > *RFP*), and GawB-Gal4 (*4G* > *RFP*) respectively. Images in (**A**,**E**,**I**,**M**,**P**,**S**) correspond to clusters of each type of neurons in brains from flies fed on the diet with 0.5% DMSO (control), or with 2 mg/mL sesamin in 0.5% DMSO (sesamin). Scale bars in (**A**,**E**,**M**,**P**) represent 10 μm, a bar in I represents 20 μm, and a bar in S represents 50 μm. (**B**,**F**,**J**) The total area showing ARE-GFP fluorescence (*n* ≥ 10). (**C**,**G**,**K**,**N**,**Q**,**T**) Ratios of the areas among ARE and RFP-positive areas in brains (*n* ≥ 4) were calculated by ImageJ for each condition. (**D**,**H**,**L**,**O**,**R**,**U**) Intensity of the GFP fluorescence was calculated by ImageJ for each condition (*n* ≥ 9). ns not significant, ** p* < 0.05, *** p* < 0.01, **** p* < 0.001, ***** p* < 0.0001, Student’s *t*-test.

**Figure 4 antioxidants-10-00924-f004:**
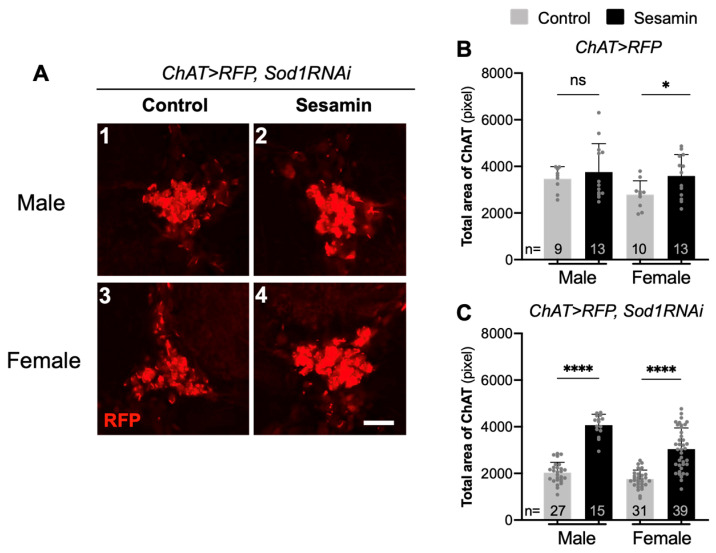
Suppression effects of sesamin on the oxidative stress-induced reduction of areas occupied by cholinergic neurons in adult brains harboring cholinergic neuron-specific depletion of *Sod1* in adult brain. (**A**) In cholinergic neurons labelled in red by RFP expression, the neuron-specific depletion of *Sod1* mRNA was carried out (*ChAT* > *RFP*, *Sod1RNAi*). Only a basal level expression of *ChAT* > *RFP* was observed in clusters of cholinergic neurons in male (**A1**) and female brains (**A3**) from flies fed on the control diet, while feeding of the diet with sesamin at 2 mg/mL enhanced the RFP fluorescence in both male (**A2**) and female brains (**A4**). (**B**,**C**) Total area occupied by the cholinergic neurons in brain (*ChAT* > *RFP*) (**B**), or in brains harboring cholinergic neuron-specific ROS accumulation due to the *Sod1* depletion *ChAT* > *RFP*, *Sod1RNAi*) (**C**) From adults (*n* ≥ 9 each). Brains were prepared from the male or female flies fed on a diet either with 0.5% DMSO only (control), or with sesamin at 2 mg/mL in 0.5% DMSO (sesamin) for 7 days. Scale bar represents 10 μm. Student’s *t*-test, ns not significant, ** p* < 0.05, ***** p* < 0.0001.

**Figure 5 antioxidants-10-00924-f005:**
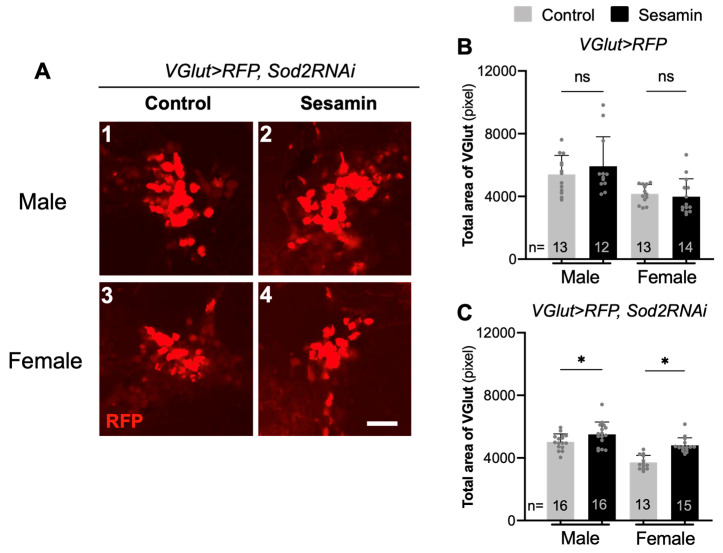
Suppression effects of sesamin on the oxidative stress-induced reduction of areas occupied by glutamatergic neurons in adult brains harboring the neuron type-specific depletion of *Sod2*. (**A**) In glutamatergic neurons labelled in red by RFP expression, the neuron-specific depletion of *Sod2* mRNA was carried out (*VGlut* > *RFP*, *Sod2RNAi*). Only a basal level expression of *VGlut* > *RFP* was observed in clusters of glutamatergic neurons in male brains (**A1**) and female brains (**A3**) from flies fed on the control diet, while feeding of the diet with sesamin at 2 mg/mL enhanced the RFP fluorescence in both male (**A2**) and female brains (**A4**). (**B**,**C**) Total area occupied by the cholinergic neurons in the brain (*VGlut* > *RFP*) (**B**), or in brains harboring glutamatergic neuron-specific ROS accumulation due to the *Sod2* depletion (*VGlut* > *RFP*, *Sod2RNAi*) (**C**), from adults (*n* ≥ 12 each). Brains were prepared from the male or female flies fed on a diet either with 0.5% DMSO only (control), or with sesamin at 2 mg/mL in 0.5% DMSO (sesamin) for 7 days. Scale bar represents 10 μm. Student’s *t*-test, ns not significant, ** p* < 0.05.

## Data Availability

The study did not report any data to public databases.

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
