# Peer review of "Sesamin Activates Nrf2/Cnc-Dependent Transcription in the Absence of Oxidative Stress in Drosophila Adult Brains"

_antioxidants, 2021, doi:10.3390/antiox10060924_

Round 1

Reviewer 1 Report

In this manuscript, Dat et al. explored the neuroprotective effect of sesamin in Drosophila. In a previous study, the authors showed that sesamin has a beneficial effect when fed to Sod1 mutant or knockdown flies that are sensitive to oxidative stress (Le et al., 2019). In this study, the authors obtained a transgenic reporter line to monitor the activation of Nrf2/CncC (ARE-GFP), a pathway that is known to mediate  anti-oxidative stress response. The authors first validate the reporter by feeding flies paraquat and confirming that this reporter is upregulated upon increased oxidative stress. Next authors showed that feeding of sesamin in the absence of oxidative stress can also induce ARE-GFP expression in a manner that is dependent on Nrf2/CncC in the brain. Overexpression of a negative regulator of Nrf2/CncC, Keap1, can also suppress the ARE-GFP expression, further supporting the involvement of this pathway downstream of sesamin application. The author further showed that ARE-GFP expression is upregulated in a subset of neuronal clusters upon sesamin feeding. Finally, the authors show that sesamin feeding can suppress what they refer to as “neuronal loss” when Sod1 or Sod2 are knocked down in a subset of neuronal clusters.

I feel the findings in this paper is interesting for the readership of Antioxidants and most experiments are well designed and conducted. I have the following major and minor points that I would like the authors to address for me to recommend this paper for publication in this journal.

Major Points:

1) The authors argue that sesamin can prevent the loss of dopaminergic, glutamatergic, and cholinergic neurons that is mediated by oxidative stress caused by knockdown of Sod1 or Sod2 in these neurons. This was based on measuring the area of these neurons by expressing RFP using neuronal cell type specific markers. Regarding this, it is not clear whether Sod1 or Sod2 knockdown causes loss of these neuronal types because the authors are not really counting the exact number of these cells. The reduction of RFP positive area may be due to loss of these neurons, or it could be because neuronal morphology has changed or other factors. If latter, this will not be considered as a model of “neuronal loss” or “neurodgeneraton”. In order for the authors to claim what they are testing here is “loss” of neurons, they should show the actual neuronal count for each neuronal subtype cluster they are examining. This can be achieved by expressing an RFP with a nuclear localization signal and by performing a counterstaining with Elav or DAPI if necessary. Also, they should specify the specific cluster of the neurons they are showing/counting. For example in their previous paper (citation [14]), they did specify which dopaminergic neurons they were quantifying (PPL1, PPM1/2, PPM3, PPL2, VUM) but in this study, the authors broadly refer to the cells they are looking at as dopaminergic. Since neuronal cluster may have variable number of cells, it is important for the authors to specify where in the brain they are looking at so others can reproduce their data.

2) In the same series of experiments, the authors assume that Sod1 or Sod2 knockdown using neuronal subtype specific markers cause oxidative stress. However, this is not shown experimentally. Upon Sod1 and Sod2 knockdown, do the author get upregulation of ARE-GFP, and is the level of ARE-GFP become further increased or not changed upon sesamin feeding? This could be done by generating flies that have Specific neuronal-GAL4, UAS-RFP, UAS-Sod1RNAi and ARE-GFP. Even if this is not possible with all GAL4s used (since construction of this genotype may be tricky if most transgenes are on the same chromosome), one can at least try this with elav-GAL4 to make sure their genetic manipulation is indeed causing oxidative stress. In addition to ARE-GFP, there are other markers of oxidative stress the authors can use in Drosophila (e.g. DHE staining, reporters reported in PMID: 22100409). The authors make strong statements including “Furthermore, we demonstrated that sesamin feeding can suppress the loss of these neurons due to excess ROS accumulation.” (line 577-579) so showing that Sod1/Sod2 knockdown indeed causes “excess ROS accumulation” that leads to “loss of these neurons (see Major Point 1)” in their hands seems critical.

Minor points

1) Early on in the paper, it wasn’t clear that CncC is the fly ortholog of Nrf2. This should be clarified the first time the authors use these terms in the introduction or abstract. For example in line 13, they can say “Nrf2 (CncC in Drosophila)” and primarily use Nrf2 throughout the remaining text. Otherwise, the text in lines 15/21 that reads “Nrf2-dependent” should be changed to “Nrf2/CncC-dependent” and keep on using both throughout the text. Also, note that in the latest version of FlyBase (http://flybase.org/reports/FBgn0262975), the official gene symbol for CncC is “cnc” and the protein symbol is “Cnc”. I feel the authors should use these official nomenclature, instead of CncC which is now considered a synonym.

2)Throughout the manuscript, the authors site their previous paper (citation [14]) to make statements like “sesamin can partially suppress the loss of dopaminergic neurons in adult brains harboring Sod1 depletion” (Lines 541-542). However, when I read this paper carefully, their Sod1 mutants only caused very minor loss of dopaminergic neurons, only causing a significant loss in two of the clusters exampled (PPL1 and VUM). Such statement will likely to be more accurate if they say “sesamin can partially suppress the loss of neurons observed in a few dopaminergic clusters in adult brains upon Sod1 depletion”

3) At the end of the abstract, the authors say “Sesamin could be used as a dietary supplement for preventing neurodegeneration associated with accumulation of oxidative stress.”. This seems to be based on the assumption that sesamin or its metabolites can cross the blood brain barrier (BBB) in human (note that the fly BBB is known to be more permeable than mammals), or some sort of signaling through the gut-brain-axis exists. Although these points are discussed in the Discussion section, they are both highly speculative. I feel the authors should tone down the statement in the abstract to something like “Sesamin could be explored as a potential dietary supplement for preventing neurodegeneration associated with accumulation of oxidative stress.”.

4) Although the authors did look at multiple neuronal types, they did not investigate the effect of sesamin in glia cells. To make the story more complete, they can try to assess ARE-GFP expression in glia cells in the presence or absence of oxidative stress using repo-GAL4 (or more glia subtype specific driver) driving UAS-RFP as a marker. Even if this is beyond the scope of this study, the authors may want to comment on the potential role of glia cells in sesamin-mediated beneficial effects.

5) Line 106, 343, Figure 3 legend etc: P{GawB}4G-GaL4 should be refered to as 4G-GAL4, I think. There are many different kind of GAL4s that are made from the P{GawB} that show very different expression patterns.

6) Lines 202-204: The authors state “These results demonstrated the activation of a Drosophila Nfr2 orthologue, CncC, in response to oxidative stress using the ARE-GFP reporter.”. However, this experiment alone doesn’t tell us whether the increase in the ARE-GFP is really due to activation of Nrf2/CncC or through alternative pathways that can somehow activate this transgene. The authors need to perform the same experiment in CncC RNAi flies, similar to what they did to show sesamin-induced ARE-GFP was dependent on CncC.

7) Lines 289-290: The authors state “These results indicated that ARE-GFP expression was stimulated especially in the brain, which was dependent on Nrf2/CncC.”. In the previous figure, the authors show that ARE-GFP expression by sesamin was seen other tissues including the gut and show body (Figure 1D, J). Were the GFP expression other than in the nervous system also suppressed by CncC RNAi upon ubiquitous knockdown or overexpression of Keap1?

8) Line 334: I am not sure what you mean by “central structure”. Please use the correct anatomical terms for the fly brain (PMID: 24559671).

9) Line 342: The author say they used Ddc-GAL4 for serotonergic neurons, but this driver stains both serotonergic and dopaminergic neurons. For specific labeling of serotonergic neurons, the authors should use Trh-GAL4 or perform a co-immunostaining with an anti-5HT antibody.

10) Line 424-425: The authors say “Next, we confirmed whether sesamin induced ChAT>RFP expression under accumulation of excess oxidative stress due to Sod1 depletion in both males and females” but this is not correct. The purpose of this experiment should not be to see if sesamin induces ChAT>RFP expression. The purpose should be to see if sesamin modulates the change in the number or morphology of cholinergic neurons, if I understand it correctly

11) Line 478-480. The authors discuss a previous paper by Bai et al. 2019 that showed that sesamin works through Nrf2, but there seems to another papers that have explored this relationship. The authors may want to consider citing and discussing Kong et al., 2016 (PMID: 27863411) and Hamada et al., 2011 PMID: 21345685)as well.

12) Line 489-490: I feel that the sentence “However, evidences regarding the oxidant activity of sesamin has not been obtained.” is not necessary here or it is out of place and belongs somewhere else.

13) Figure 4: The GFP shown here is showing a peculiar line-like pattern in Dopaminergic neurons. Do the authors know what these are (axons?), and was this pattern specific to dopaminergic neurons? Also, which dopaminergic cluster are the authors looking at here. In the previous study (citation [14]), the authors further classified the dopaminergic neurons in to PPL1, PPM1/2, PPM3, PPL2, VUM, but in this paper there is no specification.

14) Figure S3: In line 13, the term “cholinergic neurons” is used here but since this figure is only about TH-GAL4 poisitive cells, this is likely to be a mistake of “dopaminergic neurons”?

15) Grammatical errors and typo abound, mostly in the introduction. The authors should have this manuscript extensively proofread by a native speaker with descent background knowledge before the publication. Even though they add a certificate from “Editage” that this has been edited by someone in the Supplemental Material, I feel like whoever edited this paper didn’t do a good job.

Just to list a few…

Line 11: “promoted” should be “promotes”.

Line 13: “it influenced on” is weird. Something like “sesamin mediates its action through”?

Line 19: the phrase “of flies fed sesamin” should come at the end of this sentence.

Lines 70-74: neuronal types listed here should all be plural (e.g. cholinergic neurons, glutamatergic neurons…)

Line 203: “Nfr2” should be “Nrf2”

Line 206. “The organic compound” should be “This organic compound” or “DMSO”.

Line 336: “in specific neurons” should be “in specific neuronal subtypes.”

Author Response

In this manuscript, Dat et al. explored the neuroprotective effect of sesamin in Drosophila. In a previous study, the authors showed that sesamin has a beneficial effect when fed to Sod1 mutant or knockdown flies that are sensitive to oxidative stress (Le et al., 2019). In this study, the authors obtained a transgenic reporter line to monitor the activation of Nrf2/CncC (ARE-GFP), a pathway that is known to mediate anti-oxidative stress response. The authors first validate the reporter by feeding flies paraquat and confirming that this reporter is upregulated upon increased oxidative stress. Next authors showed that feeding of sesamin in the absence of oxidative stress can also induce ARE-GFP expression in a manner that is dependent on Nrf2/CncC in the brain. Overexpression of a negative regulator of Nrf2/CncC, Keap1, can also suppress the ARE-GFP expression, further supporting the involvement of this pathway downstream of sesamin application. The author further showed that ARE-GFP expression is upregulated in a subset of neuronal clusters upon sesamin feeding. Finally, the authors show that sesamin feeding can suppress what they refer to as “neuronal loss” when Sod1 or Sod2 are knocked down in a subset of neuronal clusters. I feel the findings in this paper is interesting for the readership of Antioxidants and most experiments are well designed and conducted. I have the following major and minor points that I would like the authors to address for me to recommend this paper for publication in this journal.

Major Points:

1) The authors argue that sesamin can prevent the loss of dopaminergic, glutamatergic, and cholinergic neurons that is mediated by oxidative stress caused by knockdown of Sod1 or Sod2 in these neurons. This was based on measuring the area of these neurons by expressing RFP using neuronal cell type specific markers. Regarding this, it is not clear whether Sod1 or Sod2 knockdown causes loss of these neuronal types because the authors are not really counting the exact number of these cells. The reduction of RFP positive area may be due to loss of these neurons, or it could be because neuronal morphology has changed or other factors. If latter, this will not be considered as a model of “neuronal loss” or “neurodgeneraton”.

In order for the authors to claim what they are testing here is “loss” of neurons, they should show the actual neuronal count for each neuronal subtype cluster they are examining. This can be achieved by expressing an RFP with a nuclear localization signal and by performing a counterstaining with Elav or DAPI if necessary. Also, they should specify the specific cluster of the neurons they are showing/counting. For example in their previous paper (citation [14]), they did specify which dopaminergic neurons they were quantifying (PPL1, PPM1/2, PPM3, PPL2, VUM) but in this study, the authors broadly refer to the cells they are looking at as dopaminergic. Since neuronal cluster may have variable number of cells, it is important for the authors to specify where in the brain they are looking at so others can reproduce their data.

 (Response)

We appreciate the reviewer 1’s careful reading, and valuable comments. We have previously reported that sesamin feeding partially suppressed the reduction of the neurons in several dopaminergic neuron clusters in adult brains having the neuron-specific Sod1 depletion (PMID30840390). In this study, we investigated the anti-oxidation effect of sesamin in six subtypes of neurons using a consistent protocol. In other subtypes than dopaminergic neurons, it was uncertain whether these neurons are distributed as several clusters like dopaminergic neurons, consisting of the neurons that are easy for us to distinguish. As we examined the effect of sesamin on six subtypes in this time, we selected a simpler and more sensitive method that recognizes each subtype neuron. We induced RFP using neuron-specific Gal4 drivers to label them by RFP fluorescence. However, it was sometimes not easy to count the numbers of the neurons which were close to each other. Thus, we decided to quantify the total area of RFP-positive neurons. We can understand the reviewer 1’s concern that the reduction of RFP positive area in adult brains harboring Sod1 depletion might be related to morphology change of the neurons, although it may be due to loss of these neurons. As we described in our previous paper that the loss of dopaminergic neurons occurred by Sod1 depletion or Sod2 depletion, we speculate that the reduction of the areas occupied by cholinergic neurons or glutaminergic neurons would be a consequence of the loss of the neurons, rather than the changes of neuronal morphology, or other reasons. However, we cannot exclude these possibilities at the moment. The intensity of RFP fluorescence from the neuron areas also decreased in the Sod-depleted brains. Sesamin feeding significantly restored the reduction. We interpreted the results as follows: sesamin suppressed the downregulation of cell activities including the reduced RFP expression in the neuron subtypes. Considering the reviewer1 ‘s concern, we added the following sentences that highlights the limitation of our current study in Discussion : “Considerable enhancement of ARE-dependent transcription was observed in three among five types of neurons studied, although we could not specify the clusters of dopaminergic neurons, in which the ARE-Cnc was activated by sesamin feeding in this time.” (line 547-549, page 16 in the revised manuscript with the word track). We will address the issue in next study, and present the data that would be obtained using RFPnls marker and counterstaining with anti-Elav immunostaining and DAPI staining as suggested.    

2) In the same series of experiments, the authors assume that Sod1 or Sod2 knockdown using neuronal subtype specific markers cause oxidative stress. However, this is not shown experimentally. Upon Sod1 and Sod2 knockdown, do the author get upregulation of ARE-GFP, and is the level of ARE-GFP become further increased or not changed upon sesamin feeding?

This could be done by generating flies that have Specific neuronal-GAL4, UASRFP, UAS-Sod1RNAi and ARE-GFP. Even if this is not possible with all GAL4s used (since construction of this genotype may be tricky if most transgenes are on the same chromosome), one can at least try this with elav-GAL4 to make sure their genetic manipulation is indeed causing oxidative stress.

(Response)

We appreciate the reviewer’s valuable comment and suggestions regarding a possible experiment to do. We would like to continue a study on the anti-ageing and anti-oxidation effects of sesamin. We will perform the experiments requested in our next study, and present some data which could response to the reviewer’s current concern.

In addition to ARE-GFP, there are other markers of oxidative stress the authors can use in Drosophila (e.g. DHE staining, reporters reported in PMID: 22100409).

(Response)

The DHE staining of whole brains has not sometimes worked well in our hands. It provides a higher background, an inconsistent staining even among individual brains from adults with the same genotype, and treated with the same drug administration condition. Therefore, we have often used a redox reporter, GstD1-GFP, instead of DHE staining (PMID: 30840309). Using the GFP reporter, we observed that adults with ubiquitous depletion of Sod1 showed increased the GFP fluorescence in whole bodies including heads. Moreover, the increased GFP fluorescence was suppressed after sesamin feeding for 10 days (PMID: 30840309). This observation suggests that the Sod1 depletion enhanced the ROS accumulation in the adult brains.

The authors make strong statements including “Furthermore, we demonstrated that sesamin feeding can suppress the loss of these neurons due to excess ROS accumulation.” (line 577-579) so showing that Sod1/Sod2 knockdown indeed causes “excess ROS accumulation” that leads to “loss of these neurons (see Major Point 1)” in their hands seems critical.

 (Response)

In our previous paper, we demonstrated that ectopic expression of the dsRNAs against Sod1 or Sod2 mRNA using pan-neuronal Gal4 driver, elav-Gal4 depleted the relevant endogenous mRNA efficiently. In the brains, we confirmed that the ROS accumulation was enhanced using the GstD1-GFP reporter (PMID: 25801590). We had previously shown that the depletion of Sod1 or Sod2 enhanced the loss of dopaminergic neurons in adult brains in either case (PMID: 25801590). In our another paper, we further demonstrated that the sesamin feeding performed by the same protocol mitigated the ROS accumulation in the Sod1-depletd brains (PMID: 30840309). Therefore, we consider that we investigated the sesamin’s anti-oxidation effects on cholinergic and glutaminergic neurons, in which ROS was accumulated as a consequence of neuron-specific depletion of Sod genes. However, we can understand the reviewer’s concern. Therefore, according to the reviewer’s comment, we revised the relevant sentence so as to tone down the statement as follows: “Furthermore, our current data suggested that sesamin feeding could suppress the loss of the neurons and/or their reduced activity related to excess ROS accumulation.” (line 593-595, page 17)

Minor points

1) Early on in the paper, it wasn’t clear that CncC is the fly ortholog of Nrf2. This should be clarified the first time the authors use these terms in the introduction or abstract. For example in line 13, they can say “Nrf2 (CncC in Drosophila)” and primarily use Nrf2 throughout the remaining text. Otherwise, the text in lines 15/21 that reads “Nrf2-dependent” should be changed to “Nrf2/CncCdependent” and keep on using both throughout the text.

Also, note that in the latest version of FlyBase (http://flybase.org/reports/FBgn0262975), the official gene symbol for CncC is “cnc” and the protein symbol is “Cnc”. I feel the authors should use these official nomenclature, instead of CncC which is now considered a synonym.

 (Response)

According to the reviewer’s comment, we replaced the term “Nrf2/CncC” that appeared at the first time by Nrf2 (Cnc in Drosophila) in Abstract (line 13, page 1), and replaced the first word “Nrf2” in Introduction in the same way (line 77, page 2). We also replaced the term “Nrf2” indicating a Drosophila Nrf2 orthologue by “Nrf2/Cnc” throughout (line 2, 14, 15, 17, 21, 24, 77, 80, 82, 84, 189, 194, 197, 268, 280, 290, 294, 302, 327, 332, 336, 357, 379, 385, 387, 402, 420, 481, 486, 487, 489, 492, 517, 522, 523, 556, 563, 593, 603, 612). We substituted the gene name and the protein name for cnc and Cnc, respectively (line 98, 204, 267, 283, 284, 285, 287, 288, 298, 299, 305, 310, 325, 513, 549, 596, 598, 605, 620 Figure 2A, B, C, Figure S2A, Figure S2 legend, line 13, 14, 15).

2)Throughout the manuscript, the authors site their previous paper (citation [14]) to make statements like “sesamin can partially suppress the loss of dopaminergic neurons in adult brains harboring Sod1 depletion” (Lines 541-542). However, when I read this paper carefully, their Sod1 mutants only caused very minor loss of dopaminergic neurons, only causing a significant loss in two of the clusters exampled (PPL1 and VUM). Such statement will likely to be more accurate if they say “sesamin can partially suppress the loss of neurons observed in a few dopaminergic clusters in adult brains upon Sod1 depletion”

 (Response)

We revised the sentence as suggested; “These observations are consistent with our previous findings that sesamin can partially suppress the loss of neurons in a few dopaminergic clusters in Sod1-depleted adult brains [14].” (line 415-417, page 16).

3) At the end of the abstract, the authors say “Sesamin could be used as a dietary supplement for preventing neurodegeneration associated with accumulation of oxidative stress.”. This seems to be based on the assumption that sesamin or its metabolites can cross the blood brain barrier (BBB) in human (note that the fly BBB is known to be more permeable than mammals), or some sort of signaling through the gut-brain-axis exists. Although these points are discussed in the Discussion section, they are both highly speculative. I feel the authors should tone down the statement in the abstract to something like “Sesamin could be explored as a potential dietary supplement for preventing neurodegeneration associated with accumulation of oxidative stress.”.

 (Response)

We agreed the reviewer’s concern, and revised the sentence as so to tone down the statement as follows: “Sesamin could be explored as a potential dietary supplement for preventing neurodegeneration associated with accumulation of oxidative stress.” (line 22, page 1).

Considering the reviewer’s comment, we also revised two sentences in Discussion so as to tone down the statement as follows: “Thereafter, the metabolite may be absorbed in gut epithelial cells, exported in the hemolymph, and transported toward the adult brain. If sesamin or its metabolites could cross the blood brain barrier (BBB) in Drosophila, as the Drosophila BBB is known to be more permeable than mammals[48], it could act on neurons in adult brains. Alternatively, the information of sesamin or its metabolite generated in epithelial cells may be transmitted toward the brain via a gut–brain neuronal circuit [49]. “ (line 536-542, page 16)

4) Although the authors did look at multiple neuronal types, they did not investigate the effect of sesamin in glia cells. To make the story more complete, they can try to assess ARE-GFP expression in glia cells in the presence or absence of oxidative stress using repo-GAL4 (or more glia subtype specific driver) driving UAS-RFP as a marker. Even if this is beyond the scope of this study, the authors may want to comment on the potential role of glia cells in sesamin-mediated beneficial effects.

 (Response)

We agree that a glia cell is another target candidate of sesamin, as some papers reported that glial cells can protect neurons against oxidative stress (e.g. PMID: 10349842). We believe that this is an important issue in understanding the anti-ageing activity of the chemical. We appreciate the reviewer’s comment that remind us of the importance of the involvement of glia cells. We would like to address this point in our future study. 

5) Line 106, 343, Figure 3 legend etc: P{GawB}4G-GaL4 should be refered to as 4G-GAL4, I think. There are many different kind of GAL4s that are made from the P{GawB} that show very different expression patterns.

 (Response)

We revised the abbreviation as requested, in Materials and Methods (line 107, page 3), in Results (line 344, page 9), Figure 3S, 3T, and in Figure 3 legend (392, page 12).

6) Lines 202-204: The authors state “These results demonstrated the activation of a Drosophila Nfr2 orthologue, CncC, in response to oxidative stress using the ARE-GFP reporter.”. However, this experiment alone doesn’t tell us whether the increase in the ARE-GFP is really due to activation of Nrf2/CncC or through alternative pathways that can somehow activate this transgene. The authors need to perform the same experiment in CncC RNAi flies, similar to what they did to show sesamin induced ARE-GFP was dependent on CncC.

(Response)

We agree that the observation of ARE-GFP expression alone in paraquat-fed adults is not enough to say that the increase in the ARE-GFP is due to activation of Nrf2/Cnc. However, this observation was consistent with the previously published result of the same experiment by Chatterjee and Bohmann, (2012). The authors further confirmed that the reporter expression suppressed by cncRNAi in this oxidative condition, and conversely enhanced by Keap1RNAi (PMID: 22509270). Our observation described in line 199-205 was a control experiment to confirm that the reporter expression was reproducibly activated in our hand. To investigate whether sesamin-induced activation of ARE-GFP expression in the brain depended on Cnc, we performed experiments to examine the effects of sesamin disappeared in the cncRNAi brains and showed the result in Figure 2A. We described them in section 3.3 (line 282-290). The significant activation of ARE-GFP expression failed to be observed in the brains depleted of cnc after sesamin feeding (Figure 2B, C).

7) Lines 289-290: The authors state “These results indicated that ARE-GFP expression was stimulated especially in the brain, which was dependent on Nrf2/CncC.”. In the previous figure, the authors show that ARE-GFP expression by sesamin was seen other tissues including the gut and show body (Figure 1D, J). Were the GFP expression other than in the nervous system also suppressed by CncC RNAi upon ubiquitous knockdown or overexpression of Keap1?

(Response)

Besides the GFP expression in the nervous system, we also observed a weaker signal in midgut, and quantified the levels of the GFP fluorescence in the tissues of adults harboring ubiquitous depletion of cnc (Act>cncRNAi) after sesamin feeding. We included supplementary Figure X as a separated file for reviewing. We found that the GFP expression was significantly downregulated in both males and females (ARE-GFP, Act>cncRNAi), compared with control adults (ARE-GFP, Act>+) (n>=13, p<0.0001). In contrast that sesamin feeding increased the expression in the control adults (n>=12), the GFP expression did not increase in the cnc-depleted adults after sesamin feeding (n>=9, p<0.0001). The upregulation of the expression after sesamin feeding was no longer observed in the cnc-depleted males and females. Moreover, we quantified the GFP mRNA levels in whole bodies by qRT-PCR, and confirmed that the upregulation of the mRNA levels in control flies after sesamin feeding was no longer observed in cnc-depleted flies in the same feeding condition (total RNA prepared from10-15 whole bodies (three trials) in each condition were used). From these results, we conclude that up-regulation of ARE-GFP expression in midgut is also dependent on Cnc transcription factor. We did not include these data on midgut in original manuscript, because we preferred to focus on the anti-oxidation effect of sesamin in adult brains in this time.

8) Line 334: I am not sure what you mean by “central structure”. Please use the correct anatomical terms for the fly brain (PMID: 24559671).

(Response)

According to the reviewer’s request and the fly brain, we realized that the central brain is the anatomy term corresponding to the region which we would like to describe here. Therefore, we substituted the sentence containing the improper terms as follows: “We observed strong GFP fluorescence in several areas in the central brain (Figure 1G). The mushroom body in the central brain acts as the central nervous system for olfactory learning and memory in Drosophila [37].” (line 333-335, page 9).

9) Line 342: The author say they used Ddc-GAL4 for serotonergic neurons, but this driver stains both serotonergic and dopaminergic neurons. For specific labeling of serotonergic neurons, the authors should use Trh-GAL4 or perform a co-immunostaining with an anti-5HT antibody.

(Response)

We appreciate the reviewer’s comment that reminds us the Ddc-Gal4 induces RFP in both serotonergic and dopaminergic neurons. As we do not have enough time to obtain the Trh-GAL4 stock and repeat sesamin feeding experiments in flies having serotonergic neurons labeled by Trh>RFP in this short time. However, even if Ddc-Gal4 labelled both types of neurons, our conclusion is not changed as follows: ARE-GFP expression was observed as minor spots in only a limited brain areas occupied by serotonergic and dopaminergic neurons (labeled by Ddc>RFP), and that the ARE-GFP-positive areas neither changed in amount or a fluorescence intensity after the sesamin feeding. A majority of the GFP-positive neurons among Ddc>RFP neurons may correspond to dopaminergic neurons, because more GFP signals were observed among neurons labeled by TH>RFP. Therefore, we substituted the phrase “dopaminergic and serotonergic neurons” for “dopaminergic and/or serotonergic neurons” in Abstract (line 19). We also substituted the terms “serotonergic neurons” for “serotonergic and dopaminergic neurons” in Materials and Methods (line 105, page 3), Results (line 342-343, page 9, line 373, page 10, line 381, page 10), Figure 3 legend (line 389, page 12) and Figure 3M, Discussion (line 551, page 16). We will repeat the feeding experiments and present an evidence confirming that the sesamin feeding does not provide significant alterations in serotonergic neurons in next publication.

10) Line 424-425: The authors say “Next, we confirmed whether sesamin induced ChAT>RFP expression under accumulation of excess oxidative stress due to Sod1depletion in both males and females” but this is not correct. The purpose of this experiment should not be to see if sesamin induces ChAT>RFP expression. The purpose should be to see if sesamin modulates the change in the number or morphology of cholinergic neurons, if I understand it correctly

(Response)

We agree the reviewer’s comment and revised the sentence as follows: “Next, we examined whether sesamin modulates the change in the number or morphology of cholinergic neurons under accumulation of excess oxidative stress due to Sod1-depletion in both males and females.” (line 426-428, page 13)

11) Line 478-480. The authors discuss a previous paper by Bai et al. 2019 that showed that sesamin works through Nrf2, but there seems to another papers that have explored this relationship. The authors may want to consider citing and discussing Kong et al., 2016 (PMID: 27863411) and Hamada et al., 2011 PMID: 21345685)as well.

(Response)

We checked the number 42 reference by Bai et al., 2019 again, and read other two references suggested. Accordingly, we substituted the citation as suggested (line 484, page 15).

12) Line 489-490: I feel that the sentence “However, evidences regarding the oxidant activity of sesamin has not been obtained.” is not necessary here or it is out of place and belongs somewhere else.

(Response)

As suggested, we remove the sentence (line 489-490 in the original manuscript).

13) Figure 4: The GFP shown here is showing a peculiar line-like pattern in Dopaminergic neurons. Do the authors know what these are (axons?), and was this pattern specific to dopaminergic neurons? Also, which dopaminergic cluster are the authors looking at here. In the previous study (citation [14]), the authors further classified the dopaminergic neurons in to PPL1, PPM1/2, PPM3, PPL2, VUM, but in this paper there is no specification.

(Response)

We are wondering if the reviewer would ask us about the ARE-GFP signal (Figure 3Ia’-d’) appeared in the cluster of the RFP-positive neurons. If so, we speculate that the GFP signals correspond to cell bodies of dopaminergic neurons, as they look overlapped with the RFP signals generated by TH>RFP. In our previous study, we visualized dopaminergic neurons by anti-TH immunostaining and counted the numbers of cell bodies of dopaminergic neurons (PMID:25801590). In this study, we investigated five subtypes of neurons and mushroom bodies. To carry out these experiments more quickly and more efficiently by a consistent procedure, we decided to visualize these subtypes of neurons by each neuron-specific expression of RFP using Gal4/UAS system in this time. The RFP signal was easy to recognize regions where the cell bodies existed and quantitate the amount of the neurons. However, it was sometimes difficult to count the number of the neurons. We could say that the ARE-GFP expression was induced in some clusters of dopaminergic neurons, and that the GFP fluorescence as well as the GFP-positive areas in the clusters also increased after sesamin feeding.

14) Figure S3: In line 13, the term “cholinergic neurons” is used here but since this figure is only about TH-GAL4 poisitive cells, this is likely to be a mistake of “dopaminergic neurons”?

(Response)

We revised the mistake as suggested (lines 33, 34, page 3 in Supplementary figure S3 legend). We appreciated the reviewer’s careful reading.

15) Grammatical errors and typo abound, mostly in the introduction. The authors should have this manuscript extensively proofread by a native speaker with descent background knowledge before the publication. Even though they add a certificate from “Editage” that this has been edited by someone in the Supplemental Material, I feel like whoever edited this paper didn’t do a good job.

Just to list a few…

Line 11: “promoted” should be “promotes”.

(Response)

We revised it as requested (line 11, page 1).

Line 13: “it influenced on” is weird. Something like “sesamin mediates its action through”?

(Response)

We revised it as suggested (line 13, page 1).

Line 19: the phrase “of flies fed sesamin” should come at the end of this sentence.

(Response)

We revised it as suggested (line 20, page 1).

Lines 70-74: neuronal types listed here should all be plural (e.g. cholinergic neurons, glutamatergic neurons…)

(Response)

We revised the grammatical errors as requested (lines 70, 71, 73, 74, page 2).

Line 203: “Nfr2” should be “Nrf2”

(Response)

We revised the typo as requested (line 204, page 5).

Line 206. “The organic compound” should be “This organic compound” or “DMSO”.

(Response)

We revised the phrase to “DMSO” (line 207, page 5).

Line 336: “in specific neurons” should be “in specific

neuronal subtypes.”

(Response)

We revised the phrase as requested (line 336, page 9).

Reviewer 2 Report

It is an interesting article on sesamin, in which the results could lead to new research both in mice and in other animal models of new degeneration to be able to use it with the anti-aging effects through the Nrf2-dependent pathway described and reduce oxidative stress.

Author Response

We appreciate the reviewer 2’s careful reading, and the evaluation.

Reviewer 3 Report

This is an interesting paper on a potentially important topic.  The sesame-dependent increase in Nrf2 expression and ARE-GFP are significant and an important contribution to understanding the action of sesamin.  The studies of regions of the brain and gut that are affected are also important.  What is lacking is information about the mechanism by which sesamin acts on this system.  These are detailed in the questions below.  Any information the authors can provide to answers these questions would be valuable.

Major questions:

  1. Nrf2 activation (Keap-1 oxidation) is thought to be caused by an increase in cytosolic levels of hydrogen peroxide. It would be interesting to know if sesamin reduces cytosolic levels of hydrogen peroxide. 
  2. Regarding the localization of Nrf2 activation by sesamin, does this correlate with general sensitivity of tissues to oxidation. These authors used Paraquat to test the reporter function of ARE-GFP.  Did Paraquat cause ARE-GFP expression in the same regions?
  3. In flies, what tissues take up sesamin, and how fast is it metabolized? If there is any information on this, does this correlate with the tissue-specific effectiveness of sesamin at inducing ARE-GFP?
  4. How does sesamin activate Nrf2 without causing oxidative stress? Does sesamin lead to an increase in H2O2 in cytoplasm but not mitochondria?  Does sesamin affect the interaction of Nrf2 with Keap-1?  Or does sesamin have some direct affect leading to the oxidation of Keap-1?

Minor criticism:

1.On line 233:  “Bright field microscopy” is misspelled.

Author Response

Reviewer 3

This is an interesting paper on a potentially important topic. The sesame-dependent increase in Nrf2 expression and ARE-GFP are significant and an important contribution to understanding the action of sesamin. The studies of regions of the brain and gut that are affected are also important. What is lacking is information about the mechanism by which sesamin acts on this system. These are detailed in the questions below. Any information the authors can provide to answers these questions would be valuable.

Major questions:

  1. Nrf2 activation (Keap-1 oxidation) is thought to be caused by an increase in cytosolic levels of hydrogen peroxide. It would be interesting to know if sesamin reduces cytosolic levels of hydrogen peroxide.

(Response)

We appreciate the reviewer 3’s careful reading, and valuable comments. We agree that it is quite interesting to address the issue if sesamin reduces cytosolic levels of hydrogen peroxide. In our previous study, we showed that the expression of the GstD1-GFP reporter was suppressed after the sesamin feeding (PMID: 30840309). The reporter is known to expresses GFP dependent on ROS accumulation in Drosophila (PMID: 22509270). From this previous finding, we speculate that sesamin can eventually reduce cytosolic levels of hydrogen peroxide. However, we do not have enough data on whether sesamin temporally induces ROS in Drosophila. We have an experimental plan to confirm this point by a staining of brains with another ROS indicator dye, DHE.

  1. Regarding the localization of Nrf2 activation by sesamin, does this correlate with general sensitivity of tissues to oxidation. These authors used Paraquat to test the reporter function of ARE-GFP. Did Paraquat cause ARE-GFP expression in the same regions?

(Response)

Using in vivo probe that enables us to monitor Ros accumulation in Drosophila, midgut enterocytes are identified as prominent sites of age-dependent cytosolic accumulation of hydrogen peroxide (PMID: 22100409). Generally speaking, the brain is a metabolically highly active tissue and is considered to be more susceptible to oxidative stress than any other organ (PMID: 25841340). After paraquat feeding, we observed less remarkable but significantly stronger signal of ARE-GFP expression in abdomen and head in male and female flies, and sometimes in anterior part of thorax in females (Figure 1Aa’-d’). These observations are consistent with a previous report (PMID: 22509270). However, we cannot conclude that ARE-GFP expression can be simply induced more strongly in the tissues sensitive to oxidation. Rather, when we compared the GFP fluorescence after paraquat feeding with that after sesamin feeding, we found much stronger fluorescence after sesamin feeding rather than that after paraquat feeding in the flies at the same age. We currently speculate, it is unlikely that the induction of the GFP reporter by sesamin feeding is not a direct consequence of Ros accumulation by the chemical.  

  1. In flies, what tissues take up sesamin, and how fast is it metabolized? If there is any information on this, does this correlate with the tissue-specific effectiveness of sesamin at inducing ARE-GFP?

(Response)

A previous study reported that sesamin was absorbed in human bodies with a peak plasma concentration at 5.0 h and decreased with a terminal half-life of 7.1 h (PMID: 24014208). The plasma concentration of the main metabolite, SC-1, reached a peak at 5.0 h and decreased rapidly with a terminal half-life of 2.4 h. In rats, the chemical was absorbed followed by distribution to the liver, kidney, brain, lung, and heart after oral administration (PMID: 10405978). Sesamin is absorbed efficiently and distributed over the whole body. It is highly distributed in the form of metabolites in the liver and kidney (PMID: 21345685). Consistently, our previous Drosophila study also detected the SC-1 in extracts prepared from sesamin-fed adults (PMID: 30840309). Sesamin is possibly absorbed from epithelia of somewhere in gut, mainly in midgut, although we have no direct evidences regarding tissues that take up and metabolize it in Drosophila. The kind of chemicals are generally metabolized in Malpighian tubules and fat bodies (PMID: 17488889). However, we have not observed GFP fluorescence over background level in these tissues (data not shown). We observed a weak fluorescence of the ARE-GFP reporter along midgut, and a stronger fluorescence in crop, foregut, and hindgut after sesamin feeding (Figure 1Da’-d’). We speculate that ARE-GFP expression in midgut reflected a direct effect of sesamin or its metabolites on Cnc activation. We preliminarily observed the GFP fluorescence in sesamin-fed adults every day from 2 days to 7 days after eclosion, subsequently observed the fluorescence at 10, 20, and 30 days . The GFP fluorescence was first seen in abdomen, and after that, it also appeared in heads and the signal in the regions became prominent. We have not performed a close time-course exploration neither on sesamin metabolism or ARE-GFP expression. It seems interesting to investigate whether there is a correlation between the metabolism and the tissue-specific effectiveness in activation of ARE-dependent transcription. We will challenge the issue in our next study. We appreciate the reviewer 3’s valuable comment.

  1. How does sesamin activate Nrf2 without causing oxidative stress? Does sesamin lead to an increase in H2O2 in cytoplasm but not mitochondria? Does sesamin affect the interaction of Nrf2 with Keap-1? Or does sesamin have some direct affect leading to the oxidation of Keap-1?

(Response)

It is really interesting to uncover the mechanism via which sesamin activate Nrf2. In our previous study (PMID: 30840309), we demonstrated that sesamin feeding raised mRNA levels of several anti-oxidative genes at the same time. However, we cannot conclude at present that sesamin feeding leads to an increase of hydrogen peroxide in cytoplasm in Drosophila. We showed that sesamin eventually suppressed the expression of GstD1-GFP reporter that can be induced dependent on Ros accumulation. However, we do not have enough data to conclude whether sesamin and/or its metabolite temporally induces ROS in adult bodies. A previous study using mammalian PC12 cells reported that intracellular ROS level transiently increased in the cultured cells after addition of the sesamin metabolite, SC-1 (PMID: 21345685). The authors speculated that metabolite-induced ROS not only directly interacts with SH groups of Keap1, but also activates p38 and possibly phosphorylate Nrf2. The resultant Nrf2/ARE activation may induce expression of anti-oxidant genes. Therefore, it may be reasonable to consider that the anti-oxidation effect of sesamin appears via similar processes in Drosophila. Therefore, we removed the sentence “However, evidences regarding the oxidant activity of sesamin has not been obtained.” in Discussion (original line 490, page 15), and replaced the subsequent sentence for the following sentences; “In our previous study [14], we demonstrated that sesamin feeding raised mRNA levels of several anti-oxidative genes simultaneously. And the expression of GstD1-GFP reporter that can be induced dependent on Ros accumulation was suppressed after continuous sesamin feeding. We interpreted that the suppression was a consequence of anti-oxidation effect of sesamin. By contrast, a previous study using mammalian PC12 cells reported that intracellular ROS level transiently increased in the cultured cells after addition of the sesamin metabolite, SC-1 [43]. The authors discussed that metabolite-induced ROS directly interacts with SH groups of Keap1, and activates p38 which possibly phosphorylates Nrf2. The resultant Nrf2/ARE activation may induce expression of anti-oxidant genes. Although we do not have data to conclude whether the chemical temporally induce ROS in Drosophila bodies, it may be reasonable to consider that the anti-oxidation effect of sesamin appears in Drosophila via similar processes. “

To verify the hypothesis, we would perform biochemical analysis to investigate whether SC-1 can transiently induce ROS in target tissues in which the sesamin are metabolized in our future study.  

Minor criticism:

  1. On line 233: “Bright field microscopy” is misspelled.

(Response)

We revised the typo accordingly (line 234, page 6, line 241 page 7). We appreciated the reviewer’s careful reading.

Round 2

Reviewer 1 Report

The authors made textual changes to address some of the concerns addressed by Reviewer 1 and 3 (Reviewer 2 said accept as is). For concerns/questions that requires experimental explorations, the authors mention these will be answered in future studies so they may have some time sensitive issues, which can be understandable especially under the current pandemic situation.

I am fine with most of the changes that the authors made, and I feel the new version of their text is a more accurate documentation of their data.

One thing that I still have an issue with is where the authors replaced the term "neuronal loss" with "neuronal loss and/or their reduced activity".

I am not sure if the change in the area occupied by RFP can be considered as a measure of neuronal activity. I think what this may be reflecting is "neuronal loss and/or their morphological alterations" upon oxidative stress. I will recommend the authors to consider this when finalizing the paper.

Author Response

The authors made textual changes to address some of the concerns addressed by Reviewer 1 and 3 (Reviewer 2 said accept as is). For concerns/questions that requires experimental explorations, the authors mention these will be answered in future studies so they may have some time sensitive issues, which can be understandable especially under the current pandemic situation.

I am fine with most of the changes that the authors made, and I feel the new version of their text is a more accurate documentation of their data.

One thing that I still have an issue with is where the authors replaced the term "neuronal loss" with "neuron neuronal loss and/or their morphological alterations" upon al loss and/or their reduced activity".

I am not sure if the change in the area occupied by RFP can be considered as a measure of neuronal activity. I think what this may be reflecting is "neuronal loss and/or their morphological alterations" upon oxidative stress. I will recommend the authors to consider this when finalizing the paper.

(Response)

We appreciate the reviewer’s careful reviewing again. We revised the phrase as requested (line 594-595, page 17).